# Availability and use of institutional support programs for emergency department healthcare personnel during the COVID-19 pandemic

Karin F. Hoth[1,2]*, Patrick Ten Eyck[3], Karisa K. Harland[4], Anusha Krishnadasan[5], Robert M. Rodriguez[6], Juan Carlos C. Montoy[6], Linder H. Wendt[3], William Mower[7], Kelli Wallace[4], Scott Santibañez[8], David A. Talan[4,7], Nicholas M. Mohr[4], for the Project COVERED Emergency Department Network[¶]

1 Department of Psychiatry, University of Iowa Carver College of Medicine, Iowa City, IA, United States of America, 2 Iowa Neuroscience Institute, University of Iowa, Iowa City, IA, United States of America, 3 Institute for Clinical and Translational Science, University of Iowa, Iowa City, IA, United States of America, 4 Department of Emergency Medicine, University of Iowa Carver College of Medicine, Iowa City, IA, United States of America, 5 Olive View-UCLA Education and Research Institute, Los Angeles, CA, United States of America, 6 Department of Emergency Medicine, University of California, San Francisco, San Francisco, CA, United States of America, 7 Department of Emergency Medicine, Ronald Reagan-UCLA Medical Center, Los Angeles, CA, United States of America, 8 Division of Infectious Disease Readiness and Innovation, Centers for Disease Control and Prevention (CDC), Atlanta, GA, United States of America

¶ Membership in the Project COVERED Emergency Department Network is provided in the Acknowledgments.
* karin-hoth@uiowa.edu

## Abstract

### Objectives

The COVID-19 pandemic placed health care personnel (HCP) at risk for stress, anxiety, burnout, and post-traumatic stress disorder (PTSD). To address this, hospitals developed programs to mitigate risk. The objectives of the current study were to measure the availability and use of these programs in a cohort of academic emergency departments (EDs) in the United States early in the pandemic and identify factors associated with program use.

### Methods

Cross-sectional survey of ED HCP in 21 academic EDs in 15 states between June and September 2020. Site investigators provided data on the availability of 28 programs grouped into 9 categories. Individual support programs included: financial, workload mitigation, individual COVID-19 testing, emotional (e.g., mental health hotline), and instrumental (e.g., childcare) Clinical work support programs included: COVID-19 team communication (e.g., debriefing critical incident), patient-family communication facilitation, patient services (e.g., social work, ethics consultation), and system-level exposure reduction. Participants provided corresponding data on whether they used the programs. We used generalized linear mixed models clustered on site to measure the association between demographic and facility characteristics and program use.

**Data Availability Statement:** The deidentified data upon which this manuscript is based has been

posted on Open Science Framework (osf.io) and can be accessed from the following DOI: https://osf.io/dr2kj/.

**Funding:** This project was funded by a cooperative agreement from the Centers for Disease Control and Prevention (CDC) (U01CK000480, MPI: DAT, NMM, www.cdc.gov) and the Institute for Clinical and Translational Science at the University of Iowa through a grant from the National Center for Advancing Translational Sciences at the National Institutes of Health (UL1TR002537, https://icts.uiowa.edu/). Support was additionally provided by NIH grant UM1TR004403. The sponsors played no role in the study design, data collection or analysis, decision to publish, or preparation of the manuscript. The findings and conclusions in this article are those of the authors and do not necessarily represent the views of the Centers for Disease Control and Prevention.

**Competing interests:** The authors have declared no competing interests exist.

## Results

We received 1,541 survey responses (96% response rate) from emergency physicians or advanced practice providers, nurses, and nonclinical staff. Program availability in each of the 9 categories was high (>95% of hospitals). Program use was variable, with clinical work support programs used more frequently (28–50% of eligible HCP across categories) than individual employee support programs (6–13% of eligible HCP across categories). Fifty-seven percent of respondents reported that the COVID-19 pandemic had affected their stress and anxiety, and 12% were at elevated risk for PTSD. Program use did not significantly differ for HCP who reported symptoms of anxiety and/or stress compared to those who did not.

## Conclusions

Early in the pandemic, support programs were widely available to ED HCP, but program use was low. Future work will focus on identifying barriers and facilitators to use and specific programs most likely to be effective during periods of highest occupational stress.

## Introduction

The coronavirus disease 2019 (COVID-19) pandemic created challenging work conditions across the healthcare system. Healthcare personnel (HCP) have been faced with uncertainty about resources, staffing, risk of SARS-CoV-2 exposure, concern for the well-being of coworkers and family, and providing clinical care amid visitor restrictions. Many studies have documented increases in anxiety, depression, stress disorders, and burnout among HCP since the start of the COVID-19 pandemic [1–8]. In addition to causing significant individual distress, these symptoms are associated with a higher likelihood of leaving the field of medicine [9]. Data from a public health workforce survey in 2021 suggest that just under half (44%) of workers reported that they were considering leaving their jobs within the next 5 years for retirement or other reasons, with the majority (76%) having begun to think about leaving since the start of the COVID-19 pandemic [10]. This is an alarming trend for maintaining health system capacity in the US, and information about how to address stress and burnout is urgently needed. Frontline HCP, particularly those working in emergency departments (EDs) and critical care units, have long been recognized to be at increased risk for stress and burnout and this risk was further exacerbated by the pandemic [11–15]. We previously reported prevalent symptoms of anxiety and burnout across the spectrum of ED staff during the initial wave of the pandemic and here examine support program availability and use in this at-risk population [16, 17].

Support programs for frontline HCP can mitigate the negative effects of the pandemic on HCP well-being and preserve the ability of frontline HCP to provide medical care during this public health emergency [18–26]. Research suggests that programs targeting both individual (HCP-targeted) and structural (workflow-targeted) factors are important [27]. Several types of programs have been evaluated, including education, peer support, mental health treatment, and changes to clinical workflow, with most have being evaluated in single-site studies [28–32]. Data on how hospitals implemented support programs and how they were used in the context of the COVID-19 pandemic has been lacking, however [15]. Studying the COVID-19 experience is important to understand the long-term impact of COVID-19 on health care

careers, evaluate the effectiveness of the COVID-19 health system response, and maintain readiness for future public health crises.

The goals of this study were to describe the availability and use of institutional support programs targeting both individual ED staff and clinical workflow during the early COVID-19 pandemic, and to identify HCP and facility characteristics associated with program use.

## Materials and methods

### Study design and population

We implemented this cross-sectional survey of ED personnel as a part of the COVID-19 Evaluation of Risks in Emergency Departments Project (COVERED), a prospective cohort study of emergency physicians, advanced practice providers (nurse practitioners and physician assistants), nurses and non-clinical ED personnel in 21 academic medical center EDs located in 15 states in the United States. The main study methods and results have been previously published [16, 33]. The parent study initially enrolled 1,606 participants (49.4% physicians or advanced practice providers, 25.5% nurses, and 25.0% nonclinical staff) between May 13 and August 26, 2020. For this survey, participants were from a group of HCP who had already consented to participate in the parent study. We used e-mail and text message reminders, participant incentives for completing surveys, and follow-up by local project staff as needed to ensure participation. The U.S. Centers for Disease Control and Prevention (CDC) and local institutional review boards reviewed this activity, which was conducted as public health surveillance consistent with applicable federal law and CDC policy [34]. All participants provided written informed consent, and study personnel had access to participant identifiers during the conduct of the project. This report is prepared in accordance with the Strengthening Observational Studies in Epidemiology (STROBE) statement [35].

### Survey content and administration

We administered an electronic survey regarding support program availability and use as part of the items that each participant completed in Week 4 of their enrollment in COVID-19 surveillance between June 10 and September 23, 2020. The list of programs included in the survey was developed *a priori* by study investigators based on a literature review and information provided to the research team from participating institutions (see S1 File for survey items). All site investigators and participants were asked about the same list of programs. Each site investigator reported whether their institution offered specific individual employee support programs and/or clinical work support programs within their ED at the time of data collection. On their week 4 survey, each study participant was asked whether they had personally used each of the programs on the list. For analysis and interpretation, program types were grouped into those targeting HCP individually and those targeting clinical workflow. These program groupings were created *a priori* by members of the study team based on the likely primary intended target of the program.

**Individual employee support programs.** Individual support programs included 15 types of programs that we grouped into 5 categories: (1) financial support (i.e., disability, paid time off, and additional payments); (2) workload mitigation (i.e., flexible scheduling and staff surge plans); (3) HCP COVID-19 testing (i.e., test per request and asymptomatic testing program); (4) emotional support (e.g., stress reduction/resilience training, access to mental health hotline, and social media support programs); and (5) instrumental supports (i.e., laundry services, alternative living provided, transportation, eldercare, and childcare).

**Clinical work support programs.** Clinical work support programs included 13 types of programs that we grouped into 4 categories: (1) healthcare team communication (i.e., volume

status board, personal protective equipment [36] status board, and debriefing after critical incidents); (2) family communication facilitation to mitigate social distancing visitor limits (i.e., audio-facilitated and video-facilitated); (3) patient services (i.e., interpreters, social worker, ethics consultation, and palliative care consultation); and (4) system-level HCP COVID-19 exposure reduction programs (i.e., team doffing of PPE, patient COVID-19 testing self-swabs, telehealth patient triage, telehealth patient care appointments) (S1 File).

**Stress and anxiety measures.** In the same survey, participants completed two measures rating their stress and anxiety, worded to specifically refer to the impact of the COVID-19 pandemic. First, participants responded to the following item: "In the past week, how much has the COVID-19 pandemic affected your stress or anxiety levels?" Elevated COVID-19 related stress and anxiety was defined as an answer of "somewhat" or more to this item (which corresponds to a score of $\geq 4$; see S1 File). Second, participants completed the Primary Care Post-Traumatic Stress Disorder Screen for DSM-5 (PC-PTSD-5), a 5-item screening instrument for which a score of $\geq 3$ signifies elevated risk for PTSD [37].

## Data analysis

We calculated availability for each program type (e.g., elder support, transportation) and each category of programs (e.g., instrumental supports), reporting categorical measures as counts (percentages) and continuous measures as means (standard deviations) or medians (interquartile ranges, IQRs) as appropriate. We analyzed data using SAS v.9.4 (SAS Institute, Cary, NC) and R v.4.1.1 (R Foundation for Statistical Computing, Vienna, Austria).

**Program availability and use.** We summarized the availability and use for each type of program according to the following *a priori*–defined classification plan:

- A program was considered *available* at a site if the site investigator indicated that a program was available or if 2 or more participants indicated that they had used it.

- A program was considered *used* by a participant if he/she/they indicated personal use. All participants were included in calculations for individual support program use, but only clinical staff (physicians, advanced practice providers, and nurses) were included in calculations for clinical work support program use. Program use was calculated only among sites at which that program was available.

We used this flexible definition to account for cases in which a program was available but not known to the local COVERED site investigator.

**HCP and facility characteristics associated with program use.** We used generalized linear mixed modeling (GLMM) with a logit link to make bivariate comparisons between HCP/ site characteristics (age, gender, race/ethnicity, professional role, HCP hours worked per week, HCP stress and anxiety measures, and facility COVID-19 case count the month of the survey) and support program use and included a random intercept in each model to adjust for site clustering. For each model, the GLMM provides p-values for the overall effect. Although we have multiple comparisons, we elected not to adjust our threshold for significance. We were interested in identifying hypothesis-generating factors that relate to availability and use with an emphasis on minimizing our type II rather than type I error [38].

## Results

### Participants

A total of 1,541 HCP completed the survey (response rate 96.0%), including 762 (49%) physicians or advanced practice providers, 396 (26%) nurses, and 383 (25%) non-clinical staff.

Their median age was 36 years (IQR 30–45 years). Most were female (63%) and non-Hispanic White (81%). At the time of survey completion, the median ED COVID-19 patient volume was 107 patients (IQR 57–271) per facility per month (**Table 1**).

## Support program availability and use

The proportion of sites with each type of program and each program category available and HCP use are shown in **Fig 1** and **S1 Table**. **S2 Table** presents information about which of the classification criteria resulted in coding each program as available. In 61% of cases the program was coded as available due to site investigator report and in 39% of cases the program was coded as available due to program use by ≥ 2 HCP. For HCP individual support programs, the proportion of sites that had at least one type of program available from each category was 95–100%. Individual support program category use ranged from 6% for emotional support to 13% for work demand mitigation. All sites had at least one type of program in each of the clinical work program categories in place in their ED (i.e., COVID-19 exposure reduction, patient care services, or patient–family communication facilitation and ED team communication). Clinical work support programs with the lowest availability were ethics consultations (47%) and self-administered swabs for COVID-19 patient testing (33%). Use of clinical work support programs ranged from 24% for patient–family communication facilitation to 43% for use of patient care services, which was driven by high social worker and interpreter involvement in care (24% and 36%, respectively) and low use of palliative care (7%) and ethics consultations (1%).

**Table 1. Emergency Department (ED) Health Care Personnel (HCP; N = 1,541) and facility COVID-19 volume for 21 participating COVERED EDs during the COVID-19 pandemic, June–September 2020.**

| HCP or Facility Characteristic | |
|---|---|
| Age (years), median (IQR) | 36 (30–45) |
| Hours worked per week, mean (SD) | 31.8 (10) |
| Facility COVID-19 patient volume in month before survey response, median (IQR) | 107 (57–271) |
| Gender, n (%) | |
| Female | 974 (63) |
| Male | 560 (36) |
| Transgender/Non-Conforming | 7 (1) |
| Race, n (%) | |
| White | 1214 (79) |
| Black | 110 (7) |
| Asian | 119 (8) |
| Multi/Other | 98 (6) |
| Hispanic/Latino, n (%) | 145/1499 (10) |
| Professional Role, n (%) | |
| Physician/APP | 762 (50) |
| Nurse | 396 (26) |
| Non-clinical staff | 383 (25) |
| **HCP Stress Measures, n (%)** | |
| Elevated COVID-19 related stress and anxiety rating (≥4 points) | 787/1393 (57) |
| Elevated risk for PTSD (PC-PTSD-5 ≥3 points) | 170/1398 (12) |

IQR, interquartile range; APP, advanced practice provider; PTSD, post-traumatic stress disorder; PC-PTSD-5, Primary Care Post-Traumatic Stress Disorder-5 Scale

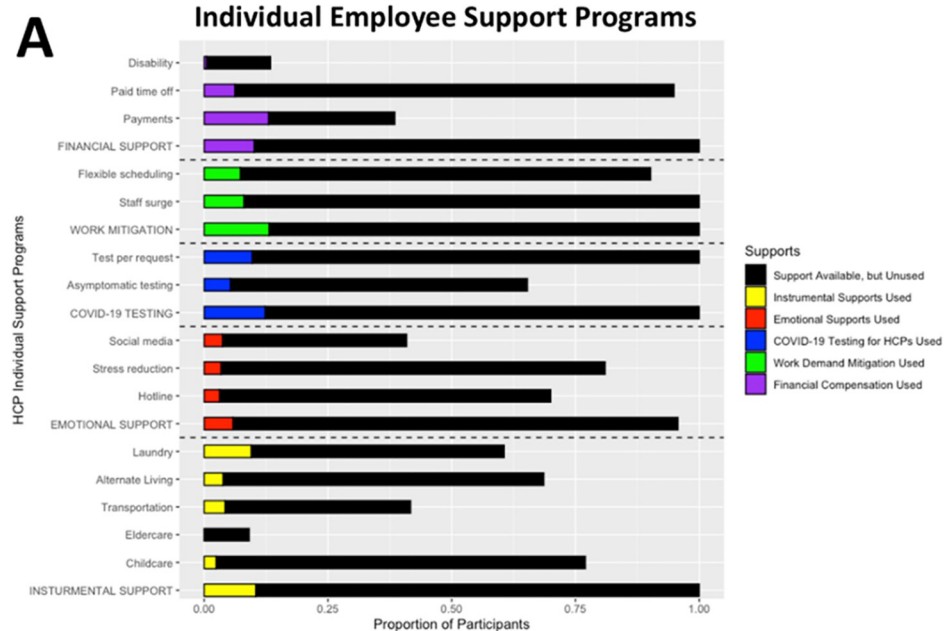

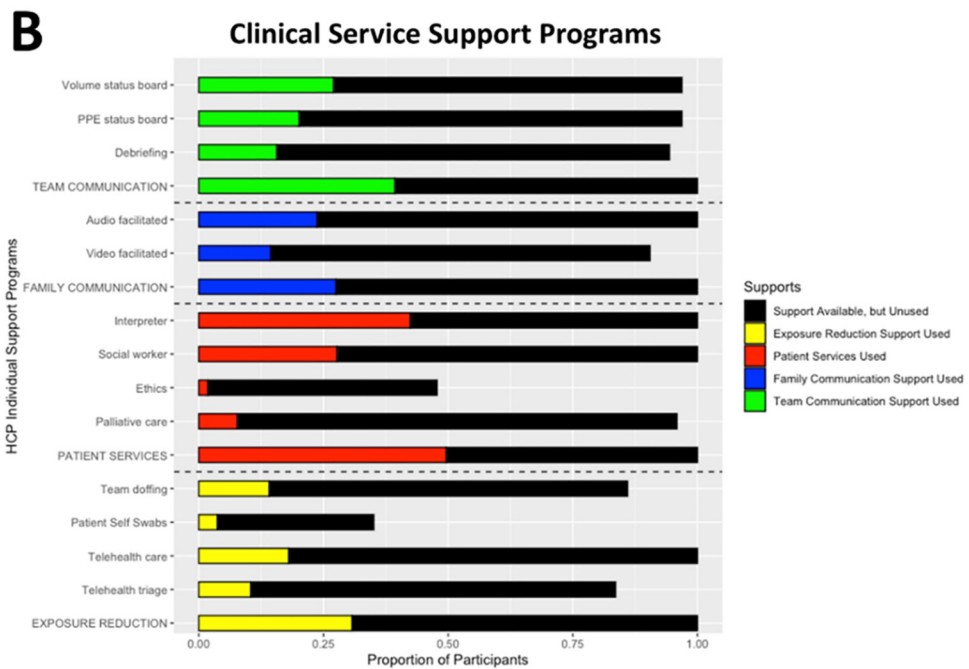

**Fig 1. Availability and use of health care personnel support programs across 21 COVERED emergency departments during the COVID-19 pandemic, June–September 2020. A.** Individual Employee Support Programs among all HCP participants, **B.** Clinical Work Support Programs among clinical HCP. The black bars indicate the proportion of participants for whom that support program type and/or category of program was available during the study period. The colored bars indicate the proportion of participants who used that support program type or category. Category pooled totals are represented in all capital letters.

Emotional support programs, such as stress reduction and mental health hotlines, were reported by site investigators to be available to 95% of participants. Overall, 6% of HCP reported using these programs at some point during the COVID-19 pandemic prior to completing the survey. In our cohort, 57% (n = 787/1393) of HCP reported elevated COVID-19–related stress or anxiety and 12% were at elevated risk of PTSD assessed via the Primary Care PTSD Screen for DSM-5 [PC-PTSD-5]; n = 170/1398). Use of emotional support programs did not significantly differ between HCP who reported elevated stress or anxiety or were at risk of PTSD based on the PC-PTSD-5 and those who were not (see **Table 2**). Use of emotional support programs was 7.5% among HCP with elevated COVID-19–related stress or anxiety and 9.3% among those who were at elevated risk for PTSD based on the PC-PTSD-5.

**HCP and facility characteristics associated with program use.** Several factors were associated with program use. Instrumental support program use differed by age, with a higher rate of use among HCPs ages 30–39 years than those younger than 30 years (13.8% vs. 8.4%; adjusted odds ratio [aOR] 1.61, 95% CI 1.01–2.56). Instrumental support programs were more likely to be used by physicians and advanced practice providers than nonclinical staff (12.1% vs. 6.8%; aOR 1.84, 95% CI 1.15–2.94). Work demand mitigation program use was most common among physicians/advanced practice providers (16.1% compared with 9.6% among nurses [aOR 1.81, 95% CI 1.23–2.67] and 10.7% among non-clinical staff [aOR 1.59, 95% CI 1.09–2.32]). Staff who worked fewer hours reported more use of work demand mitigation programs (18% for those working 20 hours per week vs. 10.2% working 40 or more hours per week; aOR 1.95, 95% CI 1.17–3.28). Physicians/advanced practice providers reported lower use of financial compensation programs than nurses (7.0% vs 16.4%; aOR 0.35, 95% CI 0.24–0.53) or non-clinical staff (9.9%; aOR 0.62, 95% CI 0.39–0.97). HCP who worked at facilities with the lowest COVID-19 patient volumes reported using financial compensation programs at the highest rate (20.5% in facilities with COVID-19 patient volume <50 in the past month vs. 10.2% in facilities with volume 50–99 [aOR 2.32, 95% CI 1.11–4.88], 5.6% in facilities with volume 100–250 [aOR 4.60, 95% CI 2.12–9.98)] and 7.4% in facilities with volume >250 [aOR 3.38, 95% CI 1.56–7.34]).

Clinical work program use was similar across HCP/facility characteristics, except that non-Hispanic White HCPs reported using programs in the patient care services category (47.8% vs. 30.6%; aOR 1.99, 95% CI 1.55–2.55) and patient-family communication category (25.7% vs. 18.7%; aOR 1.43, 95% CI 1.07–1.92) at a higher rate than HCPs from other racial/ethnic groups.

## Discussion

In this study we describe the availability and use of institutional support programs for frontline ED HCP at 21 academic hospitals in the early COVID-19 pandemic before vaccine availability and identified HCP characteristics that were associated with program use. Nearly all sites had at least one program from each category that was included in our survey (95–100%), though there were differences in availability across program types. Programs to mitigate work demands via surge staffing, paid time off, and COVID-19 testing for staff were widely available (90–100%). In contrast, elder care, disability services after SARS-CoV-2 infection, and additional payments for frontline workers were less available. Our primary observation was that despite relatively high availability of many types of individual staff support programs, HCP reported low use (<15% for each program).

Several medical professional organizations have published recommendations to improve staff support [39]. A systematic review, which included data from multiple prior virus outbreaks worldwide (i.e., severe acute respiratory syndrome [SARS], influenza caused by H1N1,

**Table 2. Demographic and facility characteristics by use of individual COVERED emergency department health care personnel support program categories during the COVID-19 pandemic, June–September 2020.** In this table, HCP use of each support program category is listed.

| | Instrumental Support | Emotional Support | COVID-19 Testing for HCPs | Word Demand Mitigation | Financial Compensation |
|---|---|---|---|---|---|
| | Used (%) | Used (%) | Used (%) | Used (%) | Used (%) |
| Age | | | | | |
| Less than 30 y | 30/359 (8.4%) | 18/336 (5.4%) | 48/359 (13.4%) | 40/359 (11.1%) | 35/359 (9.7%) |
| 30 y ≤ Age < 40 y | 83/602 (13.8%) | 43/586 (7.3%) | 82/602 (13.6%) | 89/602 (14.8%) | 66/602 (11.0%) |
| 40 y ≤ Age < 50 y | 31/308 (10.1%) | 14/297 (4.7%) | 33/308 (10.7%) | 46/308 (14.9%) | 30/308 (9.7%) |
| 50 y and Older | 17/272 (6.3%) | 11/255 (4.3) | 26/272 (9.6%) | 27/272 (9.9%) | 25/272 (9.2%) |
| Gender | | | | | |
| Male | 62/560 (11.1%) | 25/535 (4.7%) | 70/560 (12.5%) | 70/560 (12.5%) | 39/560 (7.0%) |
| Female | 98/974 (10.1%) | 61/932 (6.5%) | 117/974 (12.0%) | 131/974 (13.4%) | 114/974 (11.7%) |
| Nonbinary | 1/7 (14.3%) | 0/7 (0%) | 2/7 (28.6%) | 1/7 (14.3%) | 3/7 (42.9%) |
| Race[1] | | | | | |
| White | 123/1214 (10.1%) | 62/1157 (5.4%) | 154/1214 (12.7%) | 158/1214 (13.0%) | 123/1214 (10.1%) |
| Black/AA | 10/110 (9.1%) | 4/107 (3.7%) | 7/110 (6.4%) | 15/110 (13.6%) | 10/110 (9.1%) |
| Asian | 15/119 (12.6%) | 12/114 (10.5%) | 13/119 (10.9%) | 18/119 (15.1%) | 8/119 (6.7%) |
| Multi/Other | 13/98 (13.3%) | 8/96 (8.3%) | 15/98 (15.3%) | 11/98 (11.2%) | 15/98 (15.3%) |
| Ethnicity | | | | | |
| Hispanic/Latino | 16/145 (11.0%) | 11/145 (7.6%) | 15/145 (10.3%) | 8/145 (5.5%) | 8/145 (5.5%) |
| Non-Hispanic/Latino | 143/1354 (10.6%) | 72/1289 (5.6%) | 166/1354 (12.3%) | 191/1354 (14.1%) | 147/1354 (10.9%) |
| Professional Role | | | | | |
| Physician/APP | 92/762 (12.1%) | 52/728 (7.1%) | 103/762 (13.5%) | 123/762 (16.1%) | 53/762 (7.0%) |
| Nurse | 43/396 (10.9%) | 20/376 (5.3%) | 52/396 (13.1%) | 38/396 (9.6%) | 65/396 (16.4%) |
| Non-Clinical Staff | 26/383 (6.8%) | 14/370 (3.8%) | 34/383 (8.9%) | 41/383 (10.7%) | 38/383 (9.9%) |
| Hours Worked per Week | | | | | |
| Less than 20 h | 16/188 (8.5%) | 8/177 (4.5%) | 27/188 (14.4%) | 34/188 (18.1%) | 19/188 (10.1%) |
| 20 h ≤ Hours < 30 h | 50/390 (12.8%) | 20/371 (5.4%) | 52/390 (13.3%) | 66/390 (16.9%) | 28/390 (7.2%) |
| 30 h ≤ Hours < 40 h | 66/595 (11.1%) | 46/575 (8.0%) | 71/595 (11.9%) | 62/595 (10.4%) | 77/595 (12.9%) |
| 40 h or More | 28/343 (8.2%) | 12/328 (3.7%) | 36/343 (10.5%) | 35/343 (10.2%) | 30/343 (8.7%) |
| Elevated COVID-19 Stress/Anxiety | | | | | |
| Low stress/anxiety (0–3) | 66/606 (10.9%) | 28/577 (4.9%) | 83/606 (13.7%) | 89/606 (14.7%) | 60/606 (9.9%) |
| Elevated stress/anxiety (≥ 4) | 94/787 (11.9%) | 57/759 (7.5%) | 105/787 (13.3%) | 112/787 (14.2%) | 95/787 (12.1%) |
| Elevated risk for PTSD (PC-PTSD-5) | | | | | |
| Low risk score (0–2) | 138/1228 (11.2%) | 71/1179 (6.0%) | 165/1228 (13.4%) | 172/1228 (14.0%) | 137/1228 (11.2%) |
| High risk score (≥ 3) | 23/170 (13.5%) | 15/162 (9.3%) | 24/170 (14.1%) | 30/170 (17.6%) | 19/170 (11.2%) |
| Facility COVID-19 patient volume (in last month) | | | | | |
| Volume < 50 pts/m | 33/292 (11.3%) | 21/225 (9.3%) | 43/292 (14.7%) | 43/292 (14.7%) | 60/292 (20.5%) |
| 50 pts/m ≤ Volume < 100 pts/m | 44/413 (10.7%) | 20/413 (4.8%) | 37/413 (9.0%) | 57/413 (13.8%) | 42/413 (10.2%) |
| 100 pts/m ≤ Volume < 250 pts/m | 49/444 (11.0%) | 25/444 (5.6%) | 48/444 (10.8%) | 55/444 (12.4%) | 25/444 (5.6%) |
| 250 pts/m or more | 35/392 (8.9%) | 20/392 (5.1%) | 61/392 (15.6%) | 47/392 (12.0%) | 29/392 (7.4%) |

y, years; APP, advanced practice provider; h, hours; pts, patients; m, month

[1]The number of HCP in American Indian/Alaskan Native, Native Hawaiian/Other Pacific Islander, Multiple, and Other race groups were too small to draw meaningful inferences about race/ethnicity and thus they were grouped together for comparison.

Middle East respiratory syndrome [MERS], and Ebola), demonstrated that staff support protocols, clear communication, psychosocial interventions, and adequate infection protection were all important components of programming to improve the psychological wellbeing of medical staff during virus outbreaks [40]. Digital learning packages have been developed to aid in resiliency and coping in some medical centers [28]. Data regarding effective interventions to promote resilience in military settings and/or during natural disasters are relevant to inform programming during pandemics. For example, peer-support interventions developed initially for the military have also been used in the context of the COVID-19 pandemic [29], as peer-support has the advantage of being rapidly deployable and promoting interpersonal relationships. Indeed, in a qualitative study of HCP participating in the COVID-19 Pandemic and Emotional Well-Being Study, participants highlighted personal strategies for maintaining social connectedness that were important in coping with uncertainty from their work [41].

Our primary analyses examined rates of program use across ED HCP, although different HCP are likely to have greater needs for certain interventions than others. For this reason, additional research to target support programs optimally for those that are most likely to benefit is essential. In our study, we examined support program use comparing HCP with and without elevated stress or anxiety. Among the 57% of ED HCP who reported elevated stress or anxiety in our sample, less than 10% had engaged with an institution-provided emotional support program since the beginning of the pandemic, a rate of engagement that was similar to HCP without elevated stress or anxiety (8% vs 5%, respectively). The rate of support program use <10% should be considered in the context of the societal turmoil and work-related demands placed on ED HCP during the early period of the COVID-19 pandemic, and it suggests that employer-provided emotional support services may not reach the population of HCP most likely to be affected. The low rate of support program use highlights the importance of work to better identify social and practical barriers to institutional program use to support frontline HCP during future widespread health crises. In addition to work demands, other potential barriers may include HCP concerns about the efficacy of programs and/or concern about stigmatization by coworkers for utilizing psychosocial assistance programs.

We observed some HCP and facility characteristics that were associated with differential rates of support program use, which should be viewed as hypothesis-generating given that HCP were not asked about their perceived needs or the applicability of each program to their individual circumstances. First, HCP working over 40 hours per week reported the lowest use of individual staff support programs. Full-time staff working extra hours may not have time to engage with institutional programming, which could be one significant barrier to participation. Institutional support programming may have required additional time that was not perceived as an effective countermeasure for stress, anxiety, burnout, and PTSD by all. It is also possible that HCP who worked more hours became more familiar with use of PPE and might not have experienced the same level of concern about risk. Second, facilities with the highest COVID-19 patient volumes had the lowest proportion of HCP receiving extra financial compensation. This observation raises the possibility that in summer 2020, those facilities that were most burdened with pandemic-related demands may have had less capacity to roll out new financial support programs. This period was during the first wave of COVID-19 in most US locations and travel continued to be limited.

This study has several limitations. First, participants were ED HCP from primarily large academic and urban centers in the US with demographic characteristics generally consistent with the US hospital workforce [42]; however, their experiences may not reflect those of community and rural ED HCP. Second, information about institutional program use was collected only at one time point, and thus the cross-sectional nature of the data precludes an assessment of changes in program availability and use over time. It is possible that HCP with anxiety,

stress, burnout, and/or PTSD symptoms used program resources later in the pandemic. However, information regarding HCP needs early in the pandemic are important to prepare for the future, and the current data build on prior research from early responses during other virus outbreaks. Third, our survey asked only about availability and use of institution-based programs and did not gather information about other potential sources of support for HCP outside of work, HCP's perception of unmet program needs, the perceived or actual effectiveness of these program types, or satisfaction with programming. Fourth, our definition of program availability relied on site investigators and HCP's awareness of existing programing at their hospital and this approach likely missed some support programs that were available but unknown to either study investigators or HCP participants. HCP were not asked about their awareness of programs and instead reported only about their individual program use. We classified a program as available at a site if the site investigator indicated that it was available or if 2 or more HCP participants at the site had used it. In 39% of cases programs were coded as being available based on HCP use (i.e., instances where the site investigator had not identified the program as present). Using this flexible definition allowed us to utilize all information collected in the study regarding program availability, but the data suggest that lack of awareness of support programming is an important issue for facilities and ED departments to address. Fifth, the small sample prevented substantive analysis stratified by race/ethnicity. Future efforts can focus on improving data, including by race/ethnicity, to reflect diversity of the workforce. Finally, these data are from the initial wave of the COVID-19 pandemic from June to September 2020, a time of uncertainty about COVID-19 transmission, prior to availability of the COVID-19 vaccine, when personal protective equipment was limited at many sites in the U.S. Indeed, our data do not reflect changes in hospital programming, increasing HCP burnout, and changes in clinical practice as the pandemic has proceeded. Nonetheless, our data from frontline HCP can be used to inform our preparedness for future widespread health disasters and the impact of those events on a capable HCP workforce.

## Conclusions

In conclusion, institutional support programs for frontline ED HCP were widely available early in the COVID-19 pandemic. Despite availability and the high prevalence of symptoms of stress, anxiety, and burnout, use of individual employee support programs was low, highlighting the importance of future work to identify and address barriers to accessing current services and the need to develop more effective interventions to protect and improve ED HCP well-being during times of crisis.

## Supporting information

**S1 Table. Availability and use of institutional support programs in COVERED emergency departments, June–September 2020.**
(PDF)

**S2 Table. Coding program availability: Site investigator report of program availability and HCP reported program use.**
(PDF)

**S1 File. COVID-19 related stress, burnout and PTSD risk questionnaire.**
(PDF)

## Acknowledgments

The authors thank the COVERED participants, the participating institutions/emergency departments, and individuals. The authors acknowledge the following participating Project COVERED emergency departments: Allegheny General Hospital, Pittsburgh, PA; Baystate Medical Center, Springfield, MA; Denver Health Medical Center, Denver, CO; Detroit Receiving Hospital/Sinai–Grace Hospital, Detroit, MI; Hennepin County Medical Center, Minneapolis, MN; Jackson Memorial Hospital, Miami, FL; Johns Hopkins Medical Institute, Baltimore, MD; University Medical Center, New Orleans, LA; Mount Sinai Hospital East/Elmhurst Hospital Center, New York, NY; Orlando Regional Medical Center, Orlando, FL; University of Alabama at Birmingham Hospital, Birmingham, AL; Ronald Reagan–UCLA Medical Center/Olive View–UCLA Medical Center, Los Angeles, CA; University of Iowa, Iowa City, IA; University of Massachusetts Memorial Medical Center, Worcester, MA; University of Mississippi Medical Center, Jackson, MS; UCSF Zuckerberg San Francisco General, San Francisco, CA; UT Southwestern Medical Center/Parkland Memorial Hospital, Dallas, TX; Truman Medical Center, Kansas City, MO; Thomas Jefferson University, Philadelphia, PA; and Washington University Barnes–Jewish Hospital, St. Louis, MO.

The authors acknowledge the following individuals for their help with the study: Aishat Adeyemi, BA, Lisa Allen, MPH, MPA, Gregory Almonte, Otuwe Anya, MS, Paula Arellano-Cruz, BS, Ruzana Aronov, Danielle Beckham, RN, MSN, CRC, Lauren Buck, BS, Samuel Ceckowski, BS, Maxime Centeno, BS, Virginia, Chan, BS, Anna Marie Chang, MD, MSCE, Melissa Connor, RN, Gabriella Dashler, BS, Jenna Davis, MSN, Cynthia Delgado, BS, Veronica Delgado PA, MS, Brianna DiFronzo, Radhika L Edpuganti, BS, Alyssa Espinera, MD, Fresa Estevez, BA, Shelly Ann Evans, MBA, RN, Cathy Fairfield, BSN, Phillip Fairweather, MD, Theodore Falcon, BS, Brian Fuller, MD, MSCI, David Gallegos, BBA, Samuel Ganier, MD, Stephanie Gravitz, MPH, Jeffrey Harrison, RN, Kyle Herbert MD, Judy Hermans, MPH, Emily Hopkins, MSPH, Alan Jones, MD, Kia M. Jones, DrPH, Momina Khan, BS, Laura Iavicoli, MD, Robin Kemball, MPH, Laurie Kemble, BHSRT, CCRC, Stuart Kessler, MD, Catherine Lind RN, NP, Karina Loayza, LCSW, Carol Lynn Lyle, PA, MPH, Virginia B. Mangolds, MS, RN, Hannah Makarevich-Manilla, MPH, CCRC, Thomas Mazzocco, RN, Sarah Meram, MS, Valerie H. Mika, MS, Reynaldo Padilla, BA, Giacomo Passaglia, BS, Rebekah Peacock, BSN, Danielle Perez, BS, Kye E. Poronsky, MA, Eric Raines, EMT-P, Monica N. Ramage, MSN, RN, Kavita Rampertaap, MSN, RN, Sarah Reineck, BS, Nicole Renzi, RN, Erin P. Ricketts, MSPH, Stephanie Rodriguez, MA, Justin Sabol, BS, Valeria Samame, BA, Katie Schneider, MSN, Robert Sellman, PA, Kristine Sernulka, BSN, CCRP, Edward Sheriff, PhD, MPH, Jennifer Siller, DNP, RN, Colleen Smith, MD, Timothy Smith, BS, Patricia Slev, MD, Kelly Szabo, MPH, CCRC, Meghan Tinetti, BSN, CCRP, Denise Tritt, CIM, CRCP, Julia Vargas, BS, Samuel Vargas, BS, Kavey Vidal, BS, Lori Wilkerson, RN, CRC, Darleen Williams, DNP, APRN-CNS, Sallie Anne Wright, MPH, BSMT, and Isaias Yin, LVN. Additionally, the authors acknowledge Paul Casella, MFA for his editorial assistance.

The Project COVERED Emergency Department Network includes the following: Monica Bahamon, MPH, Jestin N. Carlson MD, MSc, Makini Chisolm-Straker, MD, MPH, Brian Driver, MD, Brett Faine, Pharm D, MS, Brian M. Fuller, MD, James Galbraith, MD, Philip A. Giordano, MD, John P. Haran, MD, PhD, Elisabeth Hesse, MD, MTM&H, Amanda Higgins, MS, Jeremiah Hinson, MD, Stacey House, MD, PhD, Ahamed H. Idris, MD, Efrat Kean, MD, Elizabeth Krebs, MD, MSc, Michael C. Kurz, MD, MS, Preeta K. Kutty, MD, MPH, Lilly Lee SM, MD, Stephen Y. Liang, MD, MPHS, Stephen C. Lim, MD, Clifford McDonald, MD, Gregory Moran, MD, William Mower, MD, PhD; Utsav Nandi, MD, MSCI, Kavitha Pathmarajah, MPH, James H. Paxton MD, Yesenia Perez, BA, Lynne D. Richardson, MD, Richard Rothman,

MD, PhD, Walter A. Schrading MD, Jessica Shuck, BA, Patricia Slev, MD, Howard A. Smith-line, MD, Michelle St. Romain, MD, Kimberly Souffront, PhD, FNP-BC, RN, Mark T. Steele, MD, Amy Stubbs, MD, Morgan B. Swanson, Josh Tiao, MD, Jesus R. Torres, MD, MPH, Stacy A. Trent MD MPH, Lisandra Uribe, BS, Arvind Venkat, MD, Gregory Volturo, MD, Kurt D. Weber, MD, and James Willey, MD. The lead author of this authorship group is Nicholas Mohr, MD (nicholas-mohr@uiowa.edu).

The findings and conclusions in this article are those of the authors and do not necessarily represent the views of the Centers for Disease Control and Prevention.

## Author Contributions

**Conceptualization:** Karin F. Hoth, David A. Talan, Nicholas M. Mohr.

**Data curation:** Karisa K. Harland, Kelli Wallace, Nicholas M. Mohr.

**Formal analysis:** William Mower.

**Funding acquisition:** Scott Santibañez, David A. Talan, Nicholas M. Mohr.

**Investigation:** Robert M. Rodriguez, Juan Carlos C. Montoy, Scott Santibañez, David A. Talan.

**Methodology:** Patrick Ten Eyck, William Mower.

**Project administration:** Karisa K. Harland, Anusha Krishnadasan, David A. Talan, Nicholas M. Mohr.

**Resources:** David A. Talan, Nicholas M. Mohr.

**Software:** Karin F. Hoth, Patrick Ten Eyck, Linder H. Wendt.

**Supervision:** Anusha Krishnadasan, Kelli Wallace.

**Writing – original draft:** Karin F. Hoth, Nicholas M. Mohr.

**Writing – review & editing:** Patrick Ten Eyck, Karisa K. Harland, Anusha Krishnadasan, Robert M. Rodriguez, Juan Carlos C. Montoy, Linder H. Wendt, William Mower, Kelli Wallace, Scott Santibañez, David A. Talan.

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
