## [Decision Letter · Decision Letter 0]

23 Aug 2023

PONE-D-23-20195Availability and Use of Institutional Support Programs for Emergency Department Healthcare Personnel During the COVID-19 PandemicPLOS ONE

Dear Dr. Hoth,

Thank you for submitting your manuscript to PLOS ONE. After careful consideration, we feel that it has merit but does not fully meet PLOS ONE’s publication criteria as it currently stands. Therefore, we invite you to submit a revised version of the manuscript that addresses the points raised during the review process.

We look forward to receiving your revised manuscript.

Kind regards,

Alireza Bornamanesh

Guest Editor

PLOS ONE

Additional Editor Comments:

Editorial Review for Manuscript PONE-D-23-20195: "Availability and Use of Institutional Support Programs for Emergency Department Healthcare Personnel During the COVID-19 Pandemic"

Editorial Recommendation: Major Revision

Review Details:

Technical Soundness and Data Support:

The reviewer emphasizes the importance of a technically sound piece of scientific research backed by robust data that adequately supports the conclusions drawn. Rigorous experimentation, including proper controls, replication, and sufficient sample sizes, is crucial. The conclusions should be aligned appropriately with the data presented.

Reviewer's Verdict: Partly

Statistical Analysis:

The reviewer expresses uncertainty regarding the adequacy and rigor of the statistical analysis performed in the manuscript.

Data Availability:

It is noted that the manuscript aligns with the PLOS Data policy's requirement for data availability. Data underlying the findings should be fully accessible, following specified guidelines.

Language and Presentation:

The reviewer underscores the importance of clear, correct, and unambiguous language in submitted articles. While PLOS ONE does not provide copyediting for accepted manuscripts, language errors should be addressed during the revision process.

General Comments:

The reviewer acknowledges important questions and response rates among those surveyed. Concerns arise regarding the limitations of the study and their potential impact on the generalizability of data and conclusions, particularly to the same population at later pandemic stages.

Suggestions are made to enhance the accessibility of the abstract by including a descriptive statement about the programs studied. Additionally, the early pandemic timepoints and short survey period are highlighted as potential factors affecting the generalizability of the responses.

In the introduction, the reviewer suggests considering references to the West Africa Ebola outbreak and military literature, offering additional context without criticism.

The reviewer poses questions about potential bias due to demographics of study volunteers and lower representation of certain groups. The alignment of enrollment percentages with broader healthcare demographics is also questioned.

The reviewer raises the possibility that increased exposure to PPE might have influenced anxiety and stress levels among healthcare personnel working extra hours.

The timing of new financial support programs in relation to the implementation of the CARES act is brought into question.

In the conclusions, the reviewer probes whether baseline stress existed in busy urban EDs prior to the pandemic, impacting the interpretation of survey responses.

Confidential to Editor:

Competing Interests:

The reviewer affirms no competing interests and highlights the topic's significance. The reviewer questions whether the study adds valuable data to the discussion, considering the response rate, timing, demographics, and limitations outlined in the manuscript.

Highlight on PLOS ONE Website:

The reviewer recommends against highlighting the submission on the PLOS ONE website based on the journal's evaluation approach.

Recognition on Web of Science Profile:

The reviewer opts for recognition of the review on their Web of Science researcher profile.

Reviewers' comments:

Reviewer's Responses to Questions

**Comments to the Author**

1. Is the manuscript technically sound, and do the data support the conclusions?

Reviewer #1: Partly

2. Has the statistical analysis been performed appropriately and rigorously? 

Reviewer #1: I Don't Know

3. Have the authors made all data underlying the findings in their manuscript fully available?

Reviewer #1: Yes

4. Is the manuscript presented in an intelligible fashion and written in standard English?

Reviewer #1: Yes

5. Review Comments to the Author

Reviewer #1: General:

- Important questions to consider and good response rates among those surveyed. There is a question whether or not the limitations of the study as highlighted by the authors make the data and conclusions less generalizable even to the same population later in the pandemic. There is also the question in my mind whether covid specific factors were being measured or baseline factors due to working in urban EDs in the US?

Abstract:

- A descriptive statement on what the programs entailed would make the abstract more accessible to the general reader.

- The early timepoints in the pandemic and short survey period call the generalizability of the responses into question.

Introduction:

- Literature from the west Africa ebola outbreak and how HCPs addressed issues similar to those raised in your manuscript might be worth commenting on citing. Military literature could also provide some interesting context. Not a criticism, just thoughts.

Methods:

Results:

- Do the authors believe there is potential bias and lack of generalizability of the data and conclusions based on the demographics of the study volunteers and lower representation of certain groups? Do the percentages enrolled in the study reflect the percentages in health care writ large?

Discussion:

"Full-time staff working extra hours may not have time"

- Is there any potential that those who worked more became more comfortable with the PPE and other processes and therefore did not experience the levels of anxiety and stress those less familiar may have experienced?

"new financial support programs"

- Had the CARES act been implemented at this point?

Conclusions:

"Despite availability and the high prevalence of symptoms of stress"

- Are we certain there was not baseline stress and burnout at busy urban EDs prior to the pandemic? The early timepoints in the pandemic and low (relative) case load beg the question of whether the survey responses were not solely associated with the pandemic. Maybe?

6. PLOS authors have the option to publish the peer review history of their article (what does this mean?). If published, this will include your full peer review and any attached files.

Reviewer #1: No

---

## [Author Response · Author response to Decision Letter 0]

9 Oct 2023

We appreciate the reviewer’s comments regarding the importance of the topic and the strength of high response rates from participants. We believe that the revised version is substantially improved. Below, we detail our responses to the major points raised in the review. 

1. “There is a question whether or not the limitations of the study as highlighted by the authors make the data and conclusions less generalizable even to the same population later in the pandemic.” 

We view the fact that data were collected early in the pandemic as both a potential strength and potential limitation. The timeframe is a strength in that the data are relevant to a period soon after a new significant outbreak of illness began, adding to prior work with past virus outbreaks. Prior authors have called attention to the need for healthcare staff support programing during prior outbreaks, yet the need persists. We have now added to the text of the discussion on lines 338-340 (i.e., “However, information regarding HCP needs early in the pandemic are important to prepare for the future, and the current data build on prior research from early responses during other virus outbreaks.”). Additionally, in response to the reviewer’s point #5 below, we have added to the text describing the needs of healthcare workers identified from past outbreaks. It is possible that the data may not generalize to later stages of the COVID pandemic, and we mention this as a limitation on lines 334-338: “information about institutional program use was collected only at one time point, and thus the cross-sectional nature of the data precludes an assessment of changes in program availability and use over time. It is possible that HCP with anxiety, stress, burnout, and/or PTSD symptoms used program resources later in the pandemic.”

2. “Do the authors believe there is potential bias and lack of generalizability of the data and conclusions based on the demographics of the study volunteers and lower representation of certain groups? Do the percentages enrolled in the study reflect the percentages in health care writ large?”

The demographic characteristics of participants in our study sample are consistent with the characteristics of hospital and outpatient clinical employees in the US in 2022, and thus we believe the data are generalizable to that population. We have added a comment about demographics of the sample and the US in general with a citation from the US Bureau of Labor Statistics. It is possible that the experience of healthcare workers in hospitals are not fully generalizable to all rural community settings, and we note this in the discussion. This text on lines 331-334 now reads: “participants were ED HCP from primarily large academic and urban centers in the US with demographic characteristics generally consistent with the US hospital workforce[42]; however, their experiences may not reflect those of community and rural ED HCP” 

3. “Are we certain there was not baseline stress and burnout at busy urban EDs prior to the pandemic? The early timepoints in the pandemic and low (relative) case load beg the question of whether the survey responses were not solely associated with the pandemic. Maybe?”

We do believe that at baseline participants in our study likely were experiencing some stress and burnout at baseline by virtue of working in an Emergency Department. Although we do not have pre-pandemic measures in the current study, we cite prior research (lines 83-88) demonstrating that Emergency Department staff are in general at elevated risk for stress and burnout compared to many other segments of healthcare. We believe that this known risk at baseline coupled with the fact that EDs early in the pandemic faced higher risks of COVID-19 exposure makes the setting an important one to focus on regarding staff support programs. 

The wording of the questions in our survey was chosen to be specific to the setting of the COVID-19 pandemic (e.g., “In the past week, how much has the COVID-19 pandemic affected your stress or anxiety levels?” “In the past week have you had nightmares related to the pandemic…”). We reasoned that this wording would reduce the likelihood of measuring baseline stress. The goal of our analyses was not to identify causes of stress or anxiety, but rather to describe the proportion of participants reporting elevated stress and anxiety at the time that we assessed support program availability. In the revised version of the manuscript, we have added text in the methods highlighting that the wording of items was specific to the pandemic (lines 155-156). 

4. Re: the abstract: “A descriptive statement on what the programs entailed would make the abstract more accessible to the general reader.” 

We agree that additional detail in the abstract would be helpful and have now added a list of the primary program types that were assessed with examples for clarification.

5. “Literature from the west Africa ebola outbreak and how HCPs addressed issues similar to those raised in your manuscript might be worth commenting on citing. Military literature could also provide some interesting context.”

Thank you for the suggestion to add information about prior outbreaks and from the context of the military. One study that provides important information from prior outbreaks, including Ebola, is a meta-analysis that we had previously cited in the discussion, but had failed to explain had spanned multiple outbreaks (e.g., SARS, MERS, H1N1, Ebola). This meta-analysis demonstrates the importance of integrating literature on healthcare center program responses over time from multiple outbreaks. Based on the reviewer’s suggestion, we now describe the data are from multiple prior outbreaks including listing the relevant viruses (lines 286-288). We also added text regarding the relevance of military literature, highlighting a COVID-19 healthcare worker peer-support program based on prior military interventions (lines 292-296).

That section of text now reads: “A systematic review, which included data from multiple prior virus outbreaks worldwide (i.e., severe acute respiratory syndrome [SARS], influenza caused by H1N1, Middle East respiratory syndrome [MERS], and Ebola), demonstrated that staff support protocols, clear communication, psychosocial interventions, and adequate infection protection were all important components of programming to improve the psychological wellbeing of medical staff during virus outbreaks [40]. Digital learning packages have been developed to aid in resiliency and coping in some medical centers [28]. Data regarding effective interventions to promote resilience in military settings and/or during natural disasters are relevant to inform programming during pandemics. For example, peer-support interventions developed initially for the military have also been used in the context of the COVID-19 pandemic[29], as peer-support has the advantage of being rapidly deployable and promoting interpersonal relationships. Indeed, in a qualitative study of HCP participating in the COVID-19 Pandemic and Emotional Well-Being Study, participants highlighted personal strategies for maintaining social connectedness that were important in coping with uncertainty from their work [41].”

6. “Is there any potential that those who worked more became more comfortable with the PPE and other processes and therefore did not experience the levels of anxiety and stress those less familiar may have experienced?”

This is an interesting alternative explanation that we had not yet considered. We have now incorporated this idea into the discussion (lines 323-325) as follows: “It is also possible that HCP who worked more hours became more familiar with use of PPE and might not have experienced the same level of concern about risk.” 

7. “Had the CARES act been implemented at this point?”

The CARES act was signed into law on March 27, 2020, which was 3 months before the start of data collection for the current study. The scope of the CARES act was wide ranging including, for example, providing support for medical equipment manufacturers, public health programs, building medical supplies for the US Strategic Stockpile, and facilitating drug applications to the US government. Effects of the CARES act could have had an impact at the healthcare institution level, and presumably any potential trickle down to the individual would have been captured via individual responses to program use.

We look forward to hearing from you regarding our revised manuscript.

---

## [Decision Letter · Decision Letter 1]

11 Jan 2024

PONE-D-23-20195R1Availability and Use of Institutional Support Programs for Emergency Department Healthcare Personnel During the COVID-19 PandemicPLOS ONE

Dear Dr. Hoth,

Thank you for submitting your manuscript to PLOS ONE. After careful consideration, we feel that it has merit but does not fully meet PLOS ONE’s publication criteria as it currently stands. Therefore, we invite you to submit a revised version of the manuscript that addresses the points raised during the review process.

We look forward to receiving your revised manuscript.

Kind regards,

Pracheth Raghuveer, MD, DNB

Academic Editor

PLOS ONE

Journal Requirements:

Reviewers' comments:

Reviewer's Responses to Questions

**Comments to the Author**

1. If the authors have adequately addressed your comments raised in a previous round of review and you feel that this manuscript is now acceptable for publication, you may indicate that here to bypass the “Comments to the Author” section, enter your conflict of interest statement in the “Confidential to Editor” section, and submit your "Accept" recommendation.

Reviewer #2: (No Response)

2. Is the manuscript technically sound, and do the data support the conclusions?

Reviewer #2: Yes

3. Has the statistical analysis been performed appropriately and rigorously? 

Reviewer #2: Yes

4. Have the authors made all data underlying the findings in their manuscript fully available?

Reviewer #2: (No Response)

5. Is the manuscript presented in an intelligible fashion and written in standard English?

Reviewer #2: Yes

6. Review Comments to the Author

Reviewer #2: The authors are to be commended for assessing this important topic regarding the availability of institutional programs mitigating risk for stress, anxiety, burnout and PTSD among healthcare professionals. This revised manuscript is generally well written with a clear objective. There are a few minor items that could use clarification.

1. On line 124 it says survey participants completed the survey in "week 4 of COVID-19 surveillance between June 10 and September 23..." Please clarify if you mean you sent the survey in the 4th week of each of those months between June and September.

2. Line 127 - 130, please clarify if participants received different surveys based on the site they worked vs a standard survey for all participants including support programs per site.

7. PLOS authors have the option to publish the peer review history of their article (what does this mean?). If published, this will include your full peer review and any attached files.

Reviewer #2: No

---

## [Author Response · Author response to Decision Letter 1]

12 Jan 2024

1. On line 124 it says survey participants completed the survey in "week 4 of COVID-19 surveillance between June 10 and September 23..." Please clarify if you mean you sent the survey in the 4th week of each of those months between June and September.

Thank you for this comment. In re-reading the section describing the survey we agree that additional clarifying detail would be helpful. The survey was sent at the 4-week mark of each individual’s participation. The survey was not sent out in one batch, but rather each participant completed the items included in this analysis are their 4-week survey time point during that broader timeframe. We have edited the survey section to more accurately reflect this.

2. Line 127 - 130, please clarify if participants received different surveys based on the site they worked vs a standard survey for all participants including support programs per site.

All participants and site investigators were asked about the same list of support programs. That list was created by the study team ahead of time. Site investigators responded regarding whether their site offered each program for staff. Participants reported about whether they had personally used each program. We have added an explicit statement that all sites received the same list of programs on the survey.

The newly edited section regarding survey methods (lines 123-135) reads as follows:

“We administered an electronic survey regarding support program availability and use as part of the items that each participant completed in Week 4 of their enrollment in COVID-19 surveillance between June 10 and September 23, 2020. The list of programs included in the survey was developed a priori by study investigators based on a literature review and information provided to the research team from participating institutions (see Appendix A for survey items). All site investigators and participants were asked about the same list of programs. Each site investigator reported whether their institution offered specific individual employee support programs and/or clinical work support programs within their ED at the time of data collection. On their week 4 survey, each study participant was asked whether they had personally used each of the programs on the list. For analysis and interpretation, program types were grouped into those targeting HCP individually and those targeting clinical workflow. These program groupings were created a priori by members of the study team based on the likely primary intended target of the program.”

---

## [Editor Report · Decision Letter 2]

31 Jan 2024

Availability and Use of Institutional Support Programs for Emergency Department Healthcare Personnel During the COVID-19 Pandemic

PONE-D-23-20195R2

Dear Dr. Hoth,

We’re pleased to inform you that your manuscript has been judged scientifically suitable for publication and will be formally accepted for publication once it meets all outstanding technical requirements.

Kind regards,

Pracheth Raghuveer, MD, DNB

Academic Editor

PLOS ONE
---

## [Editor Report · Acceptance letter]

14 Feb 2024

PONE-D-23-20195R2 

PLOS ONE

Dear Dr. Hoth, 

I'm pleased to inform you that your manuscript has been deemed suitable for publication in PLOS ONE. Congratulations! Your manuscript is now being handed over to our production team.

Kind regards, 

on behalf of

Dr. Pracheth Raghuveer 

Academic Editor

PLOS ONE